# Bi-Layered Polymer Carriers with Surface Modification by Electrospinning for Potential Wound Care Applications

**DOI:** 10.3390/pharmaceutics11120678

**Published:** 2019-12-12

**Authors:** Mirja Palo, Sophie Rönkönharju, Kairi Tiirik, Laura Viidik, Niklas Sandler, Karin Kogermann

**Affiliations:** 1Pharmaceutical Sciences Laboratory, Åbo Akademi University, Tykistökatu 6A, FI-20520 Turku, Finland; mirja.palo@abo.fi (M.P.); sjer93@gmail.com (S.R.); niklas.sandler@abo.fi (N.S.); 2Institute of Pharmacy, University of Tartu, Nooruse 1, EE-50411 Tartu, Estonia; kairitiirik@hotmail.com (K.T.); laura.viidik@ut.ee (L.V.)

**Keywords:** electrospinning, wound dressings, solvent casting, 3D printing, polymeric carrier

## Abstract

Polymeric wound dressings with advanced properties are highly preferred formulations to promote the tissue healing process in wound care. In this study, a combinational technique was investigated for the fabrication of bi-layered carriers from a blend of polyvinyl alcohol (PVA) and sodium alginate (SA). The bi-layered carriers were prepared by solvent casting in combination with two surface modification approaches: electrospinning or three-dimensional (3D) printing. The bi-layered carriers were characterized and evaluated in terms of physical, physicochemical, adhesive properties and for the safety and biological cell behavior. In addition, an initial inkjet printing trial for the incorporation of bioactive substances for drug delivery purposes was performed. The solvent cast (SC) film served as a robust base layer. The bi-layered carriers with electrospun nanofibers (NFs) as the surface layer showed improved physical durability and decreased adhesiveness compared to the SC film and bi-layered carriers with patterned 3D printed layer. Thus, these bi-layered carriers presented favorable properties for dermal use with minimal tissue damage. In addition, electrospun NFs on SC films (bi-layered SC/NF carrier) provided the best physical structure for the cell adhesion and proliferation as the highest cell viability was measured compared to the SC film and the carrier with patterned 3D printed layer (bi-layered SC/3D carrier). The surface properties of the bi-layered carriers with electrospun NFs showed great potential to be utilized in advanced technical approach with inkjet printing for the fabrication of bioactive wound dressings.

## 1. Introduction

Wound dressings with different functionalities are widely used in medical applications to aid the healing process of acute and chronic wounds [1,2]. Typically, wound dressings are designed to contribute to the inhibition of bacterial contamination and infection development, and several other pharmacological and physical protection aspects of wound healing [1,2]. In modern preparations, the carriers are made from synthetic and natural polymers that have high biocompatibility and enable localized drug delivery with improved therapeutic efficacy [1,3]. Ideally, wound dressings should provide long-term functionality, maintain a moist healing environment, promote tissue regeneration, prevent any additional damage and cause minimal inconvenience to the patient [4]. On the other hand, mechanical and adhesive properties of wound dressings determine their producibility and durability upon storage and usage [1,4]. In addition, bioactive wound dressings with biologically active ingredients and/or active pharmaceutical ingredients (APIs), such as antibiotics, anti-inflammatory agents, vitamins and growth factors, are continuously being developed [3,5].

Wound dressings are produced by different methods depending on the desired structure and the materials used. In the current study, three commonly applied methods were exploited: solvent casting, extrusion-type three-dimensional (3D) printing and electrospinning. In solvent casting, polymeric films are prepared from drying of viscous solutions of polymer(s) with/without additives and active substances in a uniformly distributed layer [6,7]. Due to the simplicity of this method, solvent casting can be time-consuming, and the properties and stability of the films are dependent on the materials used [6,7]. Nevertheless, solvent cast (SC) polymer films are structurally more robust, and highly suitable to be exploited as base layer in multi-layered formulations.

Electrospinning is a versatile technology for the fabrication of polymeric fibers with good diffusion characteristics and high surface area to volume ratio, which are beneficial for wound care applications to hinder bleeding, absorb excessive wound fluid and promote tissue regeneration [4,8,9]. The stability, reproducibility and production yield of the electrospinning process is highly dependent on the compatibility of the materials: the polymer(s), the solvent(s) and other additives or active substances [10]. Furthermore, the concept of “green electrospinning”, i.e., electrospinning of environmentally friendly and non-toxic polymeric materials and solvent systems, is gaining more attention especially in pharmaceutical and medical applications [11].

In general, 3D printing allows obtaining scaffolds with defined structures that could be exploited in various biomedical applications [12,13]. The extrusion-based 3D printing has been widely applied in bioprinting, i.e., printing of materials that contain living cells, and it is considered to be a gentle technique for manipulating semi-solid polymeric systems. This flexible method is used for tissue engineering, fabricating customized medical devices and drug delivery systems (DDSs) [14,15].

As wound dressings are expected to be actively contributing to the wound healing process, the utilization of multi-layered carriers could be beneficial. Previously, multi-layered drug-loaded formulations have been produced by electrospinning with a layer-by-layer approach [16,17,18], as well as by solvent casting [19]. Furthermore, integrated structures with improved mechanical strength can be obtained by electrospinning on top of 3D printed grid-like scaffolds [20]. A similar theoretical concept was recently presented by Maver et al. [21] for preparing a combination with carboxymethyl cellulose-based carriers manufactured by 3D printing and electrospinning for dual drug delivery from bi-layered wound dressings.

Inkjet printing is a technique, which has been used in pharmaceutical applications for the preparation of individualized DDSs [14,22]. It is a contactless method for precise deposition of liquids for tissue engineering [23] and pharmaceutical applications [24], as well as in fabrication of biosensors [25], ceramic [26] and electronic components [27], to name a few. Inkjet printing applies a drop-on-demand method for the signal-driven ejection of single droplets [28]. In DDSs, inkjet printing enables to control the precision of drug dosing and release behavior [29,30,31] and it has huge potential to be used as a method for preparing novel DDSs in combination with the polymeric carriers for wound healing applications.

Among various other materials, polyvinyl alcohol (PVA) is frequently used in polymeric carriers for medical applications due to its biocompatibility, solubility in water, non-toxicity, biodegradability, bioadhesiveness and processability [32,33]. In combination with other synthetic or natural polymers, the mechanical and physicochemical properties of PVA could be improved [34,35]. In addition, PVA-based formulations can be covalently crosslinked by different methods [36]. Sodium alginate (SA), a natural polymer extracted from brown algae, is widely used hydrophilic and biocompatible polymer in pharmaceutical applications [37]. A combination of PVA and SA is often used as a composite with improved liquid absorption, swelling, mechanical properties and thermal stability [32,38,39].

Considering the recent trends in the development of wound dressings, the utilization of combination fabrication approaches could be beneficial for preparing advanced wound dressings and novel DDSs. The aim of this study was to investigate a combinational technique for the fabrication of bi-layered carriers for the API delivery. The designed systems with modified surface layers were based on a polymer blend of PVA and SA. An electrospun layer was added onto the surface of solvent cast films of the same composition. The physicochemical, mechanical, adhesive properties of the carriers were characterized and compared with similar bi-layered carriers with 3D printed macroporous surface layer. Cell safety and viability testing was performed in order to understand the effect of different surface modifications on the cell behavior. Furthermore, a theoretical concept of inkjet printing of DDSs for the fabrication of bioactive wound dressings is presented.

## 2. Materials and Methods

### 2.1. Materials

Polyvinyl alcohol (PVA, Mowiol^®^ 20–98, Mw 125 000 g/mol, 98.0–98.8 mol% hydrolysis, Sigma-Aldrich Chemie GmbH, Steinheim, Germany) and sodium alginate (SA, Sigma-Aldrich Chemie GmbH, Steinheim, Germany) were used as film forming agents. The polymer solutions were prepared separately and mixed to obtain final solutions. The PVA solutions were obtained by dissolving PVA in purified water at 85 °C under stirring. The SA solutions were obtained by dissolving SA in purified water at room temperature (RT, 20 ± 5 °C) under stirring. For solvent casting and electrospinning, a mixture of 12% PVA and 2% SA solutions in an 80:20 volume ratio (Solution A) was used. For three-dimensional (3D) printing, a mixture of 18% PVA and 3% SA solutions in an 80:20 ratio (Solution B) was used. Thus, the polymer weight ratio was kept constant throughout the system. A corresponding physical mixture (PM) of PVA and SA was prepared for reference.

### 2.2. Preparation of Carriers

#### 2.2.1. Solvent Casting

Solvent cast (SC) films were cast from Solution A onto transparent copier film (Folex^®^IMAGING, X-10.0, Cologne, Germany) or aluminum foil with a film applicator (Multicator 411, ERICHSEN GmbH & Co. KG, Hemer, Germany) at a wet height of 500 µm. The SC films were lightly covered to prevent dusting and dried for 2 days at ambient conditions (RT and relative humidity (RH) of 35 ± 15%). The dried SC films were stored in the refrigerator (approximately 8 °C). The SC films were used as the base layer in the bi-layered carriers.

#### 2.2.2. Electrospinning

The nanofiber (NF) mats were prepared from Solution A using the eS-robot^©^ electrospinning machine (NanoNC, ESR-200Rseries, Seoul, Korea). The optimized single-spinneret electrospinning parameters were as follows: 23G needle, voltage of 11 ± 1 kV, flow rate of 0.3 mL/h and a distance of 15 cm between the needle tip and collector. The NF mats were collected onto a rotating metal collector (covered with aluminum foil) with a rotation speed of 25 rpm. The electrospinning was conducted at 25.4 ± 0.4 °C and RH of 17.5 ± 0.5%. A fiber mat of approximately 24 × 20 cm was obtained from 8 mL of Solution A. In the bi-layered solvent cast/nanofiber (SC/NF) carriers, the NFs were electrospun directly onto the SC films. Prepared NF mats and SC/NF carriers were stored in zip-lock bags in the refrigerator (approximately 8 °C) before further analysis.

#### 2.2.3. 3D Printing

The patterned 3D printed (3D) mats were obtained from Solution B onto transparent copier film with semi-solid extrusion type Biobots 1 3D printer (BioBots Inc., Philadelphia, PA, USA, currently known as Allevi). The mats were printed as a single layer (wet height: 0.15 mm) in a 40% honeycomb infill pattern with 3 perimeters, creating a macroporous mat with an area of a 1.65 × 1.65 cm square. The 3D printing was performed with a 25G needle at a pressure of 60–70 psi with a printing speed of 4 mm/s. In the bi-layered solvent cast/3D printed (SC/3D) carriers, the patterned structure was 3D printed directly onto the SC films. The bi-layered SC/3D carriers were printed with a Biobots 1 3D printer and printer System 30M (Hyrel 3D, Norcross, GA, USA). Prepared 3D printed mats and SC/3D carriers were stored in zip-lock bags in the refrigerator (approximately 8 °C) before further analysis.

#### 2.2.4. Crosslinking

A thermal crosslinking process was applied to make the carriers more stable in an aqueous environment. The thermal crosslinking was performed at 180 °C in an oven (Memmert, DIN 12880, Schwabach, Germany) for 10 min.

### 2.3. Characterization Methods

#### 2.3.1. Visualization

Microscopic images of the carriers were obtained with an Evos XL Imaging System (InvitrogenTM, Thermo Fisher Scientific, Waltham, MA, USA), ProScope digital microscope (Bodelin Technologies, PSEDU-100, Oregon City, OR, USA) and scanning electron microscopes (SEM). The non-crosslinked electrospun NF mats were visualized with SEM (EVO MA 15, Zeiss^®^, Oberkochen, Germany) after magnetron-sputter coating with a 3 nm gold layer in an argon atmosphere. The crosslinked NF mats and bi-layered carriers were visualized with SEM (Leo Gemini 1530, Zeiss^®^, Oberkochen, Germany) that was equipped with secondary electron and In-Lens detectors. The bi-layered carriers were sputter-coated with carbon using a vacuum evaporator prior to imaging. The images were analyzed with ImageJ software (1.51j8, National Institute of Health, Bethesda, MD, USA).

#### 2.3.2. Texture Analysis

The mechanical strength of the carriers was measured by a puncture test method using TA.XT*plus* Texture Analyzer (Stable Micro Systems, Surrey, UK) equipped with a film support rig and a spherical stainless steel probe (⌀ = 5 mm). The measurements settings were as follows: pre-test speed of 2 mm/s, test speed of 1 mm/s and a post-test speed of 10 mm/s. The maximum force (N) needed to break the carriers was recorded (burst strength).

A digital caliper (Mitutoyo, 500-171-21, CD-6”, Kawasaki, Japan) was used to measure the thickness of the carriers.

#### 2.3.3. Solid-State Characterization

The infrared (IR) spectra were collected from the carriers and the raw materials with a universal attenuated total reflectance Fourier transform IR spectroscope (ATR-FTIR, UATR Two, Perkin Elmer, Llantrisant, UK). The measurements were conducted in a spectral range from 450 to 4000 cm^−1^ with 4 scans per spectrum (*n* = 3). The data collection and the baseline correction of the IR spectra were performed with Spectrum 10.03 software (PerkinElmer, Llantrisant, UK).

The thermal properties of the SC films and electrospun NFs were measured by differential scanning calorimetry (DSC, Pyris Diamond, PerkinElmer, Waltham, MA, USA). Samples of 1–3 mg were analyzed in 30 µL aluminum pans with pierced lids. A heating rate of 10 °C/min was used in a measuring range of 25–250 °C. An N_2_ purge gas was used with a flow rate of 40 mL/min. The DSC system was calibrated using indium (156.6 °C). Thermograms were baseline corrected prior to analysis.

#### 2.3.4. Stability Study

A short-term stability study for one month was performed with non-crosslinked and thermally crosslinked SC films and electrospun NF mats. The samples were stored at two separate conditions: i) RT and low humidity (RH of 0%), and ii) accelerated conditions [40] at elevated temperature (40 °C) and humidity (RH of 75%). The solid-state properties of the samples were measured at three time points (24 h, 1 week, 1 month) by ATR-FTIR and DSC spectroscopy.

### 2.4. Behavior of Bi-Layered Carriers in Biorelevant Conditions and During DDSs Preparation

#### 2.4.1. Swelling and Degradation in Aqueous Environment

The stability of the non-crosslinked and crosslinked carriers in aqueous environment was studied. The samples (1.65 × 1.65 cm squares) were weighed and immersed in 10 mL of pH 7.4 phosphate buffer saline (PBS) solution. The samples were mixed (30 rpm) at 37 °C for 24 h, 3 days and 7 days. The mass of the samples was recorded with a microbalance (d = 1 µg, MYA 2.4Y, Radwag Wagi Elektronicze, Radom, Poland).

The swelling degree was calculated as percentage of swelling ratio using Equation (1) [41]:Swelling degree (%) = (*W*_s_ − *W*_0_ / *W*_0_) × 100,(1)
where *W*_s_ is the swollen sample and *W*_0_ is the initial sample weight. The swollen samples were weighed after excess fluid had been removed with filter paper immediately after taking the samples out of the PBS solution.

The degree of degradation was calculated as percentage of weight loss by Equation (2) [41]:Degradation degree (%) = (*W*_0_ − *W*_1_) / *W*_0_ × 100,(2)
where *W*_0_ is the initial weight of the sample and *W*_1_ is the dry weight of the sample obtained after PBS immersion. The samples were dried for 7 days under a ventilated fume hood prior to weighing.

#### 2.4.2. Simulated Bioadhesion Study

The adhesion of the non-crosslinked and thermally crosslinked carriers to artificial skin (VitroSkin^®^ N-19, IMS Inc., Cape Coral, FL, USA) was tested with TA.XT*plus* Texture Analyser (Stable Micro Systems, Surrey, UK) equipped with a mucoadhesion rig and a cylinder delrin^®^ probe (⌀ = 10 mm) at ambient conditions. A method developed by Tamm et al. [42] was used in a slightly modified format. Shortly, circular samples (⌀ = 11 mm) of the carriers were prepared and attached to the probe with an adhesive double-sided tape (Scotch™, 3M, Livonia, MI, USA). Simulated wound fluid (200 µL/sample) was pipetted on the artificial skin before the measurement. The simulated wound fluid contained 2% bovine serum albumin (Sigma-Aldrich, St Louis, MO, USA), 0.02 M CaCl_2_·2H_2_O (Merck, Darmstadt, Germany), 0.4 M NaCl (Sigma-Aldrich, St Louis, MO, USA), 0.08 M tris(hydroxymethyl)-aminomethane (Merck, Darmstadt, Germany) and purified water [43]. The testing conditions were set as follows: pre-test speed 0.5 mm/s, test speed 0.5 mm/s, post-test speed 5 mm/s, applied force 1 N, return distance 100 mm, contact time 60 s, and trigger force 0.05 N. Scotch™ adhesive double-sided tape and commercial wound dressing Aquacel™ (ConvaTec Inc., Reading, UK) were used as references.

#### 2.4.3. Safety of Bi-Layered Carriers and the Effect of Surface Modification on Cell Viability

Safety studies of bi-layered crosslinked and non-crosslinked carriers were carried out using baby hamster kidney (BHK-21) fibroblast cells. Cells were grown in the Glasgow Minimal Essential Medium (GMEM) supplemented with 7.5% fetal bovine serum (FBS), 2% Tryptose Phosphate Broth Difco (TPB, Midland Scientific Inc., Omaha, NE, USA), 20 nM HEPES, 100 µg/mL penicillin and 100 µg/mL streptomycin. For the viability study, cells were placed into 24-well plates and grown 24 h at 37 °C in 5% CO_2_ incubator. Samples with a size of 1 cm^2^ were placed into the wells on top of the cells, media was changed and incubated for another 24 h. Safety was assessed qualitatively by optical microscopy and quantitatively by trypan blue exclusion (automated cell counter, Invitrogen, Thermo Fischer Scientific, Waltham, MA, USA), counting dead and live cells from which the viability (%) was calculated. The experiment was carried out in triplicate, whereas cells exposed only to the growth medium or placed on top of the glass plate in growth medium were used as healthy untreated controls.

To evaluate the viability of cells on bi-layered crosslinked carriers and crosslinked SC film, the MTS cell proliferation assay was performed. Samples were placed into 24-well plates using cell crown inserts (CellCrown^®^, Scaffdex Oy, Tampere, Finland). Size of the samples was 1.5 × 1.5 cm. 500 µL of BHK-21 cells were seeded on the samples, the number of cells per well was approximately 50,000. 700 µL of GMEM was added. After 24 h incubation the samples were removed from the inserts, washed with 1× PBS, then transferred to 500 µL Dulbecco’s Modified Eagle medium (DMEM) without phenol red, and 80 µL of MTS reagent (K300-500, Biovision, Milpitas, CA, USA) was added. After 45 min of incubation at 37 °C in 5% CO_2_ incubator the colored medium was pipetted onto a 96-well plate and absorbance was measured using plate reader at 490 nm. The experiment was carried out in triplicate together with appropriate controls. The graphs show the relative viable cell numbers whilst the substrates with the highest cell numbers obtained were considered as 100%.

#### 2.4.4. Surface Behavior During Inkjet Printing

Inkjet printing was used to investigate the surface behavior of the bi-layered carriers upon contact with aqueous ink solution. A mixture of propylene glycol (PG, Sigma-Aldrich, St Louis, MO, USA) and purified water in 40:60 ratio with viscosity of 3.9 mPa·s and surface tension of 47.4 mN/m was used as the ink solution. Red food color (9%, Dr. Oetker Sverige AB, Gothenburg, Sweden) was included in the ink solution prior to printing for better visualization. The ink was deposited on the carriers with a piezoelectric inkjet printer (PixDro LP50, Meyer Burger Technology Ltd., Thun, Switzerland) equipped with a Spectra^®^ SL-128AA printhead (Fujifilm, Valhalla, NY, USA) at a resolution of 100 dpi. The average droplet size of the ink during printing was approximately 45 pL.

A CAM 200 contact angle goniometer (KSV Instruments Ltd., Espoo, Finland, currently known as Biolin Scientific) paired with a camera (Basler, Ahrensburg, Germany) and OneAttension software (Theta1.4) was used for contact angle (sessile drop method) measurements. The shape of the 5 µL droplets was recorded at 24 ± 1 °C in the triplicate measurements.

### 2.5. Data Analysis

Results are expressed as a mean ± standard deviation (SD). Statistical analysis was performed by two-tailed Student’s *t*-test assuming unequal variances with Microsoft Office Excel 365 ProPlus software (*p* < 0.05) where applicable.

## 3. Results and discussion

### 3.1. Characterization of Solvent Casted (SC) Films, Electrospun Nanofibers (NFs) and 3D Printed Mats

Before the preparation of bi-layered carriers, each layer was prepared separately and characterized. The SC films obtained were transparent with smooth surface (Appendix A). After thermal cross-linking the films turned to yellowish, but no other structural changes nor cracking were observed (Appendix A).

The SEM imaging showed the formation of well-defined nanofibrous structures by electrospinning (Figure 1). Electrospinning process was slow, but the solution was electrospinnable within those conditions after optimization. A low degree of merging of fibers was noted in non-crosslinked NF mats, possibly due to imperfect drying of the polymer solution. However, no visible changes were detected in the fiber morphology after crosslinking (Figure 1C). The diameter of the fibers followed a unimodal distribution trend in the nanometer-range (Figure 1). The average diameter increased slightly after crosslinking with a significant change in the distribution range of the fiber diameter.

The 3D printed mats were prepared by semi-solid extrusion 3D printing. Recently, this method was applied to prepare warfarin-loaded orodispersible films [44]. Here, the 3D printing of macroporous mats was performed onto a plastic support liner with a honeycomb infill pattern. The 3D printed pattern was clearly visible after printing (Appendix A) and the dried mats remained intact after removal from the copier film. The 3D printed mats contained a patterned matrix with lemon-shaped pores with dimensions of approximately 990 × 1620 µm (*n* = 8).

### 3.2. Preparation and Structure of Bi-Layered Carriers

The bi-layered carriers were prepared through multi-step manufacturing processes (Figure 2). Adding a surface layer onto the SC film allows modifying the structure and functionality of wound dressings. In the bi-layered carriers, the use of same polymer composition enabled to create structures, where a thin layer of electrospun NFs adhered to the SC film base layer.

Preparing uniformly fibrous scaffolds for skin regeneration and wound healing requires a production of fiber mats that have the thickness and mechanical strength suitable for application handling. The production speed of electrospun fibers can be slow and varies considerably depending on the electrospinning setup and polymer system. The presented approach addresses these key aspects in the production of wound dressings. Adding a thin layer of electrospun fibers onto a strong polymer film could decrease the production time, improve cost-effectiveness, while still providing the unique properties, e.g., nanofibrous and porous structure together with the required adhesive and skin protective properties.

Even though, the 3D printed mats could be used without any support layer, the bi-layered SC/3D carriers were prepared to investigate the differences presented by the additional surface layer onto the SC base layer. The patterned 3D printed mat covered 76 ± 4% (*n* = 2) of the sample area leaving a macroporous structure with small palpable cavities on the carrier surface. The shape of the design was not retained entirely, and the printing coverage was higher than the theoretical estimate (51%). This can be explained by the insufficient viscosity of the printed polymer solution, and the high shear forces applied in the needle tip during printing that further affected the viscosity of the solution.

In the bi-layered SC/NF and SC/3D carriers, the directly applied additional layer merged with the SC film and was non-removable. The structure of the carriers remained intact and the layers did not separate from each other (no lamination) during thermal crosslinking. In all formulations, the thermal treatment resulted in extensive crosslinking that caused a visible color change from white to yellowish as also previously reported [41,45]. In a study by Fuchs at al. [46], heat sealing was utilized to unite polycaprolactone (PCL)-based SC films and 3D structures obtained by melt electrospinning writing. This type of an extra step was not required in the setup presented here for the polymer blend with PVA and SA. It gives some supportive evidence that also drug-loaded systems (mats) can be produced using the same approach: the combination of the two techniques and using same ingredients.

### 3.3. Physical Properties

The physical properties of the SC films and the bi-layered carriers were measured to evaluate the effect of the additional layer to the SC base layer (Table 1). Expectedly, the thickness of the bi-layered carriers was increased compared to the SC film. The crosslinking affected the thickness of bi-layered SC/NF carriers, suggesting a notable thermal expansion in the fibrous structure. The thickness of both crosslinked bi-layered formulations was comparable to the copy paper (0.09 ± 0.01 mm).

The puncture test revealed a high deviation in the burst strength of the carriers (Table 1). This can be contributed to the non-uniformity in the polymer films due to fluctuations in the drying conditions, volume of residual solvent [47] as well as other preparation related factors. The difference in the burst strength (N) between non-crosslinked and thermally crosslinked samples was shown to be statistically non-significant. However, the decrease in the distance at break (mm) in the crosslinked SC film and the bi-layered carriers suggests that the thermal treatment affected the elasticity of the formulation. On the other hand, the SC/NF carriers showed a slight increase in the average burst strength after crosslinking and longer distance to break (mm) compared to the SC/3D carriers. The addition of 3D printed layer seemed to decrease the elasticity of the crosslinked carriers, whereas this effect was not pronounced for the crosslinked carriers with electrospun NFs. Previously, it has been reported that NF mats from only PVA become brittle after thermal crosslinking [41]. In this study, the addition of SA to the mixture seemed to improve the integrity of the NF layer, and thus durable carriers with improved mechanical properties were obtained.

The effect of crosslinking was demonstrated in the stability study in an aqueous environment. The non-crosslinked samples disintegrated rapidly after immersion into the PBS solution (<24 h). The thermally treated samples remained visibly intact throughout the study period of 7 days. In an earlier study, crosslinked films of PVA and SA were investigated for 48 h with 10 subsequent cycles of dissolution and drying without any visible changes to their integrity [48]. Nevertheless, the swelling and degradation processes occurred simultaneously at a constantly changing ratio in all crosslinked formulations.

The crosslinked SC films and SC/NF carriers showed no significant weight loss after 7 days (Table 1). Whereas, the average degradation degree for SC/3D samples was 2.8% after 7 days, indicating that the additional 3D printed layer contributed to the interaction with the buffer solution due to increased surface area and/or decreased degree of crosslinking (thickness and volume of the carrier was higher compared to the SC film). As a comparison, the macroporous structure of the patterned 3D printed mats gave rise to an approximately 7% decrease in weight after immersion in the PBS solution for 7 days.

Due to the use of same materials, the combined layers in the bi-layered carriers showed high compatibility. No layer separation was detected in the bi-layered formulations after immersion in the PBS solution (Figure 3). Furthermore, the distinct surface structures remained visible after 7 days in aqueous environment. Notably, the surface of the SC/NF carriers was smoother due to the swelling and adhesion of the NFs to each other (Figure 3B).

An initial burst in the uptake of water and/or salts from the buffer solution manifested within the first 24 h and declined later. The swelling degree was significantly higher for the SC/NF carriers compared to the SC/3D carriers (Table 1). The absorptive properties attributed to the NF structures promotes their applicability in wound care applications [9,49]. For example, manyfold higher absorption ratios have been reported for fibrous alginate wound dressings and other commercial gauzes using a different experimental setup [50]. In PVA/SA hydrogels, the water absorption and swelling capacity has shown to be dependent on the SA content [51]. Thus, the results suggest that in the preparation of bi-layered carriers the incorporation of NFs improves the degree of liquid uptake and usability as wound dressings. Furthermore, the liquid absorption degree would be improved by adjusting the polymer ratio in the formulations.

### 3.4. Stability and Solid-State Characterization

The stability and solid-state characteristics of SC films and electrospun NF mats were studied at two different conditions for one month. Spectroscopic analysis identified several absorbance bands characteristic to the raw materials (Figure 4). The spectrum of SA displayed absorbance bands in the fingerprint region for carboxylate group at 1598 and 1407 cm^−1^, and skeletal C–O–C linkage at 1027 cm^−1^ [52,53]. The high degree of hydrolysis (98.0–98.8%) for pure PVA was seen by the low intensity of the absorption band in the 1700–1750 cm^−1^ region [54]. The preparation of aqueous solutions for electrospinning and 3D printing as well as the preparation processes themselves affected slightly the PVA hydrolysis degree in the prepared formulations. The intensity decrease of absorbance band at approximately 1715 cm^−1^ was observed in the spectra of the non-crosslinked/crosslinked SC films and electrospun NF mats compared to the spectrum of PVA and SA physical mixture (Figure 4). Interestingly, the crosslinking of the SC films and NF mats did not have an additional effect on the hydrolysis degree of PVA (based on the intensity of the absorbance band at 1715 cm^−1^). Hence, the intensity of the band at 1715 cm^−1^ was similar with the intensity in the spectra of non-crosslinked samples (NF mats and SC films).

A broad absorbance band, attributed mainly to the hydroxyl groups in the molecules, was present in the range of 3600–3200 cm^−1^. The broadening and intensity changes of the absorbance band for OH^−^ groups were in correlation with the water content, degree of hydrolysis and the crystallization degree of the formulations. The ratio between the intensities of the absorbance bands at 3400–3200 cm^−1^ and 1420 cm^−1^ is related to the degree of chemical crosslinking [54]. The decreased ratio in the intensities refers to the higher degree of crosslinking. In the crosslinked SC films and electrospun NF mats it was seen that the ratio of the mentioned peaks was decreased. Thus, a relationship between thermal crosslinking and the presence of OH^−^ groups was confirmed. These changes are most likely due to the decrease in the water content and hydrolysis. However, the presence of other crosslinking mechanisms cannot be confirmed.

The changes related to the absorption band at 1141 cm^−1^ (marked with an asterisk on Figure 4 and Figure 5) were evaluated closely due to the strong relation between the peak intensity and the crystallinity of PVA [41,54,55]. All NF mats prepared by electrospinning were in an amorphous state, whereas the SC films showed to exhibit a semi-crystalline state (Figure 4). The amorphous state of the non-crosslinked NF mats was stable during storage at low humidity (Figure 5). However, the degree of crystallinity in the non-crosslinked NF mats increased at elevated storage conditions (40 °C and RH of 75%). The crystallinity in the non-crosslinked SC films increased over time at both storage conditions.

Physical crosslinking of the SC films and electrospun NF mats resulted in the crystallization of the formulations. Similar findings have been previously reported by Miraftab et al. [41], where it was concluded that the degree of crystallinity is dependent on the temperature and heating time. The broadening of the absorption band for the hydroxyl group indicated that the crosslinking decreased the residual water content and/or inter- and intramolecular hydrogen bonding [56]. This phenomenon shows correlation with the physical properties of the carriers. During the stability study, no apparent changes in the solid-state properties of the crosslinked samples were detected at both storage conditions. On the other hand, the changes in the absorbance band for OH^-^ groups indicated that the non-crosslinked SC films were more affected by the storage conditions than other formulations. The intensity of the absorbance band at 3400–3200 cm^−1^ increased about 70% after one month at elevated conditions (40 °C and RH of 75%) and decreased about 40% after storage at RT and RH of 0%.

The changes seen in the analysis of the infrared spectra are supported by the results from the thermal analysis by DSC. The semi-crystalline PVA exhibited a melting endotherm at 217 °C (ΔH ≈ 75 J/g), which is in accordance with previous results [57]. SA is an amorphous material that decomposes in two steps—dehydration (approximately 100 °C) and exothermic decomposition (240–260 °C) [58]. The melting endotherm of non-crosslinked electrospun NFs (219 °C) increased slightly after incubation at elevated storage conditions due to changes caused by high humidity and temperature. As mentioned earlier, no changes were observed in the crosslinked electrospun NF mats at both storage conditions. The melting endotherm remained at approximately 214 °C with high enthalpy values (ΔH ≈ 70 J/g) characteristic for formulations with crystalline PVA. The thermal analysis revealed no significant changes in the melting endotherm of SC films (217 °C) throughout the stability study. Interestingly, the SC samples exhibited some impurities and initial degradation above melting temperature, which was not apparent in the electrospun NF mats (data not shown).

A correlation between the solid-state properties and physical properties is obvious, and the properties of the carriers are highly affected by the thermal crosslinking. Therefore, the effect of time and storage-dependent solid-state changes on the physical properties of the carriers should be further investigated.

### 3.5. Bioadhesion within Simulated Wound Fluid

The force (N) and work (N·mm) required for the detachment of the carriers from the wetted artificial skin are presented in Figure 6. The effect of crosslinking on the adhesive properties is clearly visible for the samples. The NF mats and bi-layered carriers showed a statistically significant decrease in the detachment force (N) and the work of adhesion (N·mm) after crosslinking. However, the intact structure of the crosslinking patterned 3D printed mats seemed to increase their adhesion, while the polymer structure in the non-crosslinked samples dissolved fast upon contact with the simulated wound fluid.

In the bi-layered SC/NF carriers, the adhesiveness of the non-crosslinked formulations was comparable to the SC film base layer, indicating that the thin nanofibrous layer dissolved rapidly and did not contribute to the adhesion process. The relatively low adhesiveness of this formulation could be related to the low polymer concentration and viscosity of the in situ formed hydrogel [59].

The analysis revealed that in the crosslinked bi-layered SC/NF carriers the addition of electrospun NFs decreased the adhesiveness significantly compared to the SC film and SC/3D carriers (Figure 6). This indicates that bi-layered SC/NF carriers would be easily removable from damaged tissue and suitable for wound care applications. The intact crosslinked electrospun NF layer decreased the contact with the artificial skin surface most likely due to extensive swelling [60] with detachment values close to the detection limit. The SC/3D printed carriers behaved similarly to the SC film, suggesting that the macroporous surface structure is not enough to obtain favorable adhesive properties required for wound care.

An ideal wound dressing should be non-adherent and cause no additional injuries for the wound and pain to the patient during the removal [61]. However, in some cases the ability of wound dressings/skin adhesives to adhere on the skin or even on the damaged skin (e.g., hydrocolloid dressings) is crucial [62]. In the latter case, a good interaction between the dressing and the skin contributes to the absorption of excessive wound fluid and promotes healing by maintaining a moist environment [63]. Ideally, the wound dressings should be self-adhesive to the wound surface, yet easy and painless to remove [42]. Polymeric wound dressings are widely used; besides, if crosslinking is used to enhance their robustness, the adhesive properties of the formulations might be weakened [64,65], which was also apparent in the current study. Furthermore, the incorporation of APIs and/or use of polymer blends in the electrospun NF mats might significantly affect their bioadhesion, as it has been demonstrated with polyvinylpyrrolidone [42]. Thus, a balance between the mechanical and adhesive properties should be aimed for.

### 3.6. Safety of Bi-Layered Carriers and the Effect of Surface Modification on Cell Viability

Safety of all carriers were confirmed using BHK-21 fibroblast cells and direct viability testing method. All materials were biocompatible, and no statistically significant differences were observed between the carriers (non-crosslinked and thermally crosslinked) and untreated controls (Figure 7A and Appendix A).

It was also confirmed that no changes in cell numbers were present due to the thermal crosslinking of the carriers. The latter nicely supports the findings observed in the solid-state characterization, as most likely no harmful substances (e.g., substances cytotoxic to fibroblast cells) were produced within the materials as a result of high temperature. Hence, such carriers, also stated as generally regarded as safe (GRAS) materials and approved within several medical devices by the U.S. Food and Drug Administration (FDA) [66], could be used for further wound dressing development.

Effect of surface modification on the cell viability was also determined. Crosslinked bi-layered SC/NF carrier enhanced the cell adhesion and proliferation as the highest cell viability (statistically significant, *p* < 0.05) was measured compared to other carriers, i.e., SC film and SC/3D carrier (Figure 7B). Interestingly, the cell viability on SC film and 3D/SC carriers was approximately the same (Figure 7B). Both materials (SA and PVA) are known to be biocompatible, biodegradable and good for the cells [67,68]. However, it is known that cells are able to sense the environment [69] and in addition to the material properties (internal chemical composition and the mechanical properties) [70], the material surface modification may change the behavior of the cells [71,72]. The overall idea of using surface modified structures is to stimulate the cells or their behavior. Usually the carriers used for the treatment of wounds are designed according to specific requirements and should provide a highly porous structure with interconnected pores [73]. These characteristics provide cells an appropriate environment for growth. Thus, carriers with such properties act as physical substrates for cell adhesion, proliferation, and differentiation, as well as for the integration to the host tissue in order to regenerate the defects, in case the mats are used as skin substrates. In the present study, it was also revealed that the porous electrospun surface layer provided the best surface for the cells. Interestingly, patterned 3D printed layer did not show superior cell viability values compared to the non-porous SC film. The latter most likely is due to the effect of too large pore size of the 3D printed layer. It is known that the best cell attachment for fibroblasts takes place between the pore size of 50–160 µm and nano-pored scaffolds with 1 µm pores improve the initial cell-surface interactions the most [74]. The pore size as well as geometry affect the successful regeneration of tissues [74].

### 3.7. Surface Properties During Inkjet Printing

Prior to inkjet printing, contact angle measurements were performed on the SC films and the bi-layered carriers (Figure 8). The contact angle of macroscale (5 µL) ink droplets on the non-crosslinked SC film was in the range of 31°–36°, which was significantly lower compared to the contact angle of purified water (70.8° ± 2.5°) on the same SC film.

The crosslinking decreased the contact angle of the solution on the SC films (Figure 8). The ink droplets on the non-crosslinked and crosslinked SC/NF carriers spread out and absorbed the ink droplet quickly (<60 s). Thus, the contact angle could not be determined for these carriers. Whereas, the bi-layered SC/3D carriers behaved somewhat similarly to the SC films, which was expected. Although, the grid-like printed pattern on the SC/3D carriers caused higher variability in the contact angle values and irregularity in the droplet shape depending on the location of the deposition.

In general, the hydrophilicity of the carriers and the rheological properties of the ink solution contributed to the spreading and absorption of the solution into the surface layer of carriers. On the other hand, the carriers remained intact due to the thermal crosslinking.

Thereafter, inkjet printing was utilized to deposit microscale droplets of aqueous ink solution onto the bi-layered carriers to further investigate their surface characteristics. These measurements allowed revealing the behavior of the carriers and get an understanding about their use as DDSs for the wound healing application. The SEM images visualize clearly how the liquid adsorption differs on the surfaces of different carriers (Figure 9 and Figure 10). The shape of the dried ink droplets is dependent on the liquid-carrier physical interaction, but also on the drying process. The higher contact angle between the liquid and the carrier surface allows obtaining deposited droplets with well-defined shape. Whereas, in the formulations that exhibit a lower contact angle the ink spreads out faster, and thus the shape of the droplet is uneven as the drying process is occurring simultaneously.

On the SC/3D carriers, the ink droplets solidified mainly on the top layer and some sedimentation from the ink additives is visible on the droplet edges (Figure 10). The shape of the droplets indicates that the ink absorbed into the surface layer of the non-crosslinked SC/3D carrier, whereas on the crosslinked counterpart the ink dried in steps with minor penetration into the surface. In such formulations, the deposition of large liquid volumes would favor the droplets to merge within each other and result in an uneven coverage of the carrier.

The uniformity of ink deposition, which is a necessity to produce precision printed systems, was achieved for SC/NF carriers (Figure 9). In non-crosslinked SC/NF carriers, the ink solution dissolved the electrospun nanofibrous surface layer, resulting in a uniform liquid distribution. Besides that, the SC support film allowed for the carrier to remain intact and easy to handle. In crosslinked SC/NF carriers, separate ink droplets were not detectable by SEM, suggesting that the small amount of the solvent evaporated rapidly from the voids of the fibrous structure and any solid additives from the ink adhered to the surface of the NFs. The change in the light contrast (as difference in conductivity) of the SEM images obtained with the InLens detector could be related to this phenomenon (Figure 9).

Since the integrity of the nanofibrous system was maintained, the liquid absorption in the crosslinked SC/NF carriers is firstly dependent on the packing density of the NFs, and secondly on the swelling capacity of the crosslinked polymer(s). However, the defined liquid deposition is most likely dependent on the thickness of the NF layer. The hypothesis is that above a certain level, the higher liquid volume could result in an irregular spreading of the solution due to contact with the smooth SC base layer, as was noticed for the SC/3D samples.

The use of these polymeric bi-layered carriers could be beneficial for delivering APIs through the dermal or buccal administration ways. In these DDSs, the APIs can be incorporated by inkjet printing [29,75]. The precision of liquid dosing and location of solution deposition provided by inkjet printing could be highly beneficial for increasing the efficacy-dose ratio in antimicrobial preparations [30,76]. In the current study, the successful printing of a drug-free base solution demonstrates that bi-layered carriers with suitable nanofibrous surface layer could be useful components in the printed DDSs. The behavior of active ingredients in this type of printed DDSs is highly dependent on the drug properties and their physicochemical interactions with the polymer carriers. Thus, potential wound dressings with additional bioactive functionality should be studied case by case.

## 4. Conclusions

The utilization of bi-layered carriers allows adjusting the three main aspects of polymeric carriers: physical properties, bioadhesion properties and cell–carrier interactions. Crosslinked bi-layered carriers with a solvent cast (SC) base layer and an electrospun nanofibrous surface (SC/NF carrier) were prepared with good stability and physical properties. Adding a porous layer on the mechanically strong SC films improved their absorption capacity and suitability for wound care applications. The bi-layered carriers were non-adherent in the simulated bioadhesion study, presenting favorable properties for dermal use (non-adherent dressings) with minimized damage to the skin upon removal. All prepared carriers (non-crosslinked and crosslinked) were safe and exhibited good biocompatibility towards BHK-21 fibroblast cells. Surface modification by electrospinning (bi-layered SC/NF carrier) increased the cell viability compared to the SC film and carrier with pattern 3D printed layer (bi-layered SC/3D carrier). Hence, bi-layered SC/NF carriers are good physical substrates for the cells and provide help in skin regeneration during wound healing.

The inkjet printing trial demonstrated a uniform liquid absorption profile in the bi-layered carriers with electrospun nanofibers. Furthermore, the surface modification provides a platform for the fabrication of advanced drug delivery systems with active substances (e.g., pharmaceutical and/or bioactive substances) for enhanced wound healing or drug delivery through various administration routes.

Further investigation aims at developing multilayered wound dressings with bioactive substances and in more biorelevant conditions. The relationship between the properties of the carriers and the composition of the nanofibrous layer will be evaluated and the physical properties and stability of advanced drug delivery systems will be studied in depth. In addition, biodegradability and specific bioactivity behavior testing will be carried out to assess the applicability of bi-layered carriers with drug substances for clinical use.

## Figures and Tables

**Figure 1 pharmaceutics-11-00678-f001:**
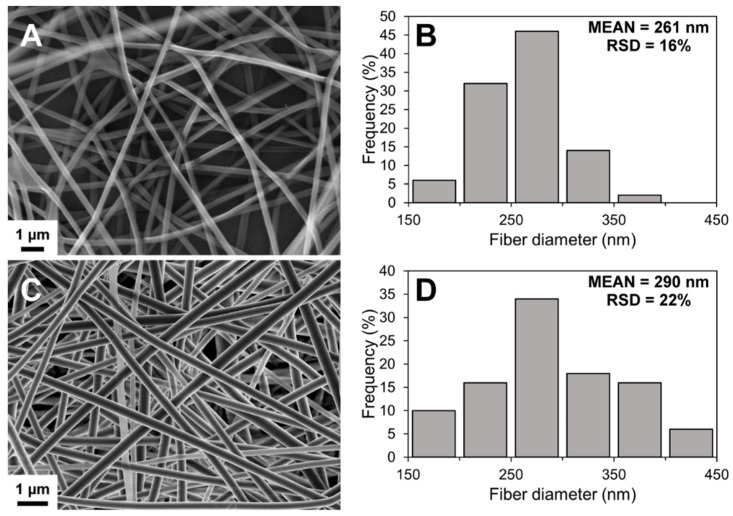
Scanning electron microscopy (SEM) image (**A**) and fiber diameter histogram (**B**) of non-crosslinked solvent cast/nanofiber (SC/NF) carriers, and SEM image (**C**) and fiber diameter histogram (**D**) of crosslinked SC/NF carriers with mean and relative standard deviation (RSD) values (*n* = 50).

**Figure 2 pharmaceutics-11-00678-f002:**
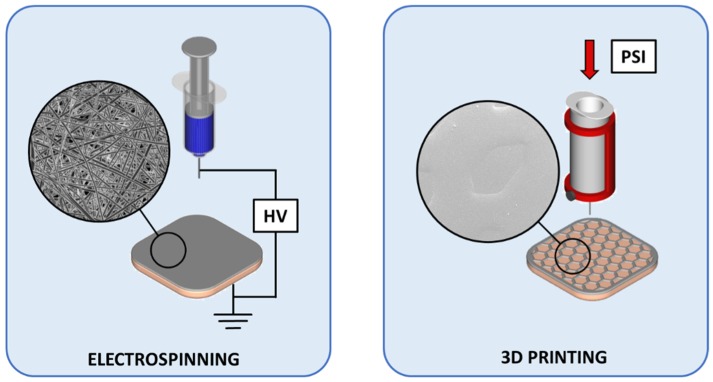
Preparation schematics of the designed bi-layered carriers for wound care. Key: HV—high voltage; 3D—three dimensional; PSI—pressure in pound per square inch.

**Figure 3 pharmaceutics-11-00678-f003:**
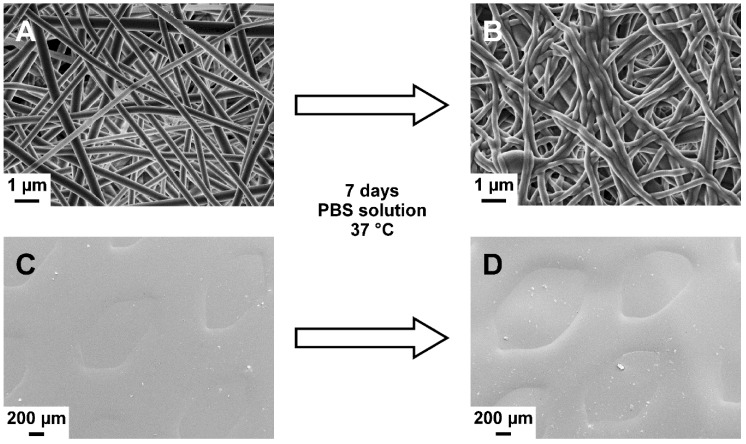
Scanning electron microscopy (SEM) images of crosslinked solvent cast/nanofiber (SC/NF) (**A**,**B**) and solvent cast/3D printed (SC/3D) (**C**,**D**) carriers before (**A**,**C**) and after (**B**,**D**) immersion in phosphate-buffered saline (PBS) solution at 37 °C for 7 days.

**Figure 4 pharmaceutics-11-00678-f004:**
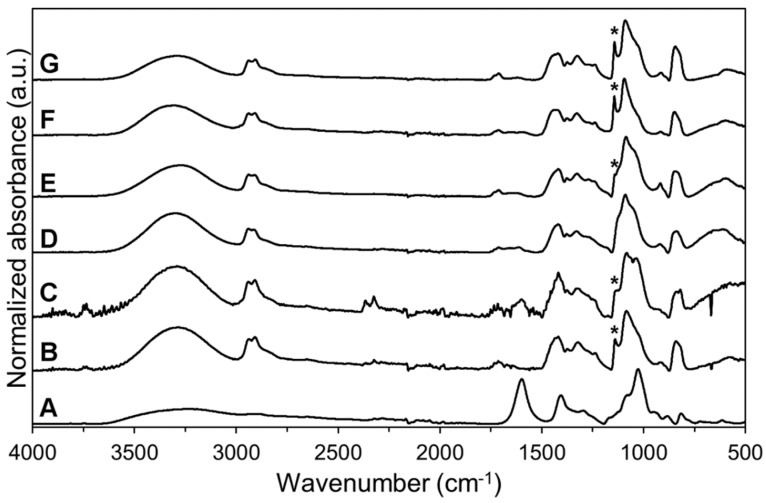
Attenuated total reflectance Fourier transform IR spectroscopy (ATR-FTIR) spectra (spectral range from 4000 to 500 cm^−1^) of sodium alginate (SA) (**A**) and polyvinyl alcohol (PVA) powders (**B**); physical mixture of SA and PVA powders (**C**); non-crosslinked electrospun nanofiber (NF) mat (**D**); non-crosslinked solvent cast (SC) film (**E**); crosslinked electrospun NF mat (**F**) and crosslinked SC film (**G**). Asterisk (*) denotes the absorption band at 1141 cm^−1^ used to evaluate the crystallinity of PVA.

**Figure 5 pharmaceutics-11-00678-f005:**
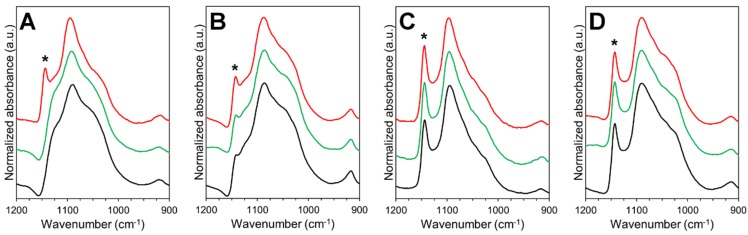
ATR-FTIR spectra (spectral range from 1200 to 900 cm^−1^) from short-term stability study of non-crosslinked electrospun nanofiber (NF) mat (**A**) and solvent cast (SC) film (**B**); crosslinked electrospun NF mat (**C**) and SC film (**D**). Color legend: black—initial state, green—1 month at room temperature (RT) and relative humidity (RH) of 0%, red—1 month at 40 °C and RH of 75%. Asterisk (*) denotes the absorption band at 1141 cm^−1^ used to evaluate the crystallinity of polyvinyl alcohol (PVA).

**Figure 6 pharmaceutics-11-00678-f006:**
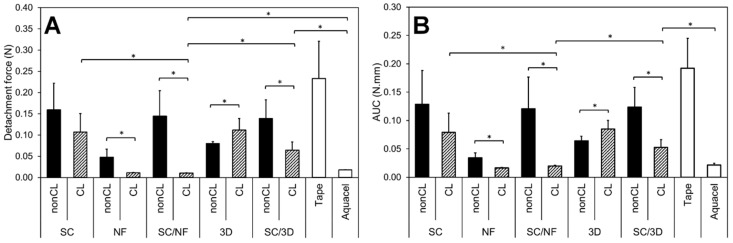
The detachment force (N) of adhesion (**A**) and work of adhesion as area under the curve (AUC, N.mm) (**B**) of non-crosslinked (nonCL) and crosslinked (CL) solvent cast (SC) film, electrospun nanofiber (NF) mat, bi-layered solvent cast/nanofiber (SC/NF) carrier, patterned 3D printed (3D) mat and solvent cast/3D printed (SC/3D) carrier as well as tape and Aquacel^®^ wound dressing as reference (*n* = 5–8). Asterisk (*) denotes the statistical difference with *p* < 0.05.

**Figure 7 pharmaceutics-11-00678-f007:**
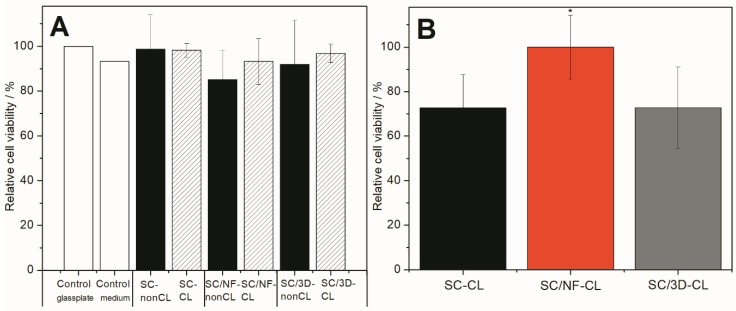
Cell viability assay results from trypan blue exclusion test (**A**) and MTS test (**B**) on different carriers: non-crosslinked (nonCL) and crosslinked (CL) solvent cast (SC) film, solvent cast/nanofiber (SC/NF) carrier and solvent cast/3D printed (SC/3D) carrier. Cells were incubated for 24 h and analyzed. Results presented as relative cell viability (%). In direct assay with trypan blue exclusion testing, the cell number of untreated BHK-21 fibroblasts on glass-plate was considered as a positive control providing 100% cell viability (other positive control was cells in plastic wells in growth medium). In MTS tests, the carrier with the highest cell viability (SC/NF-CL carrier) was considered as 100%. Data are expressed as mean ± standard deviation (*n* = 3). Asterisk (*) denotes the statistical difference with *p* < 0.05.

**Figure 8 pharmaceutics-11-00678-f008:**
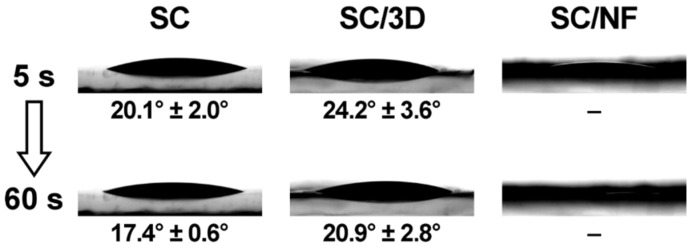
Contact angle (°) of the ink solution on crosslinked solvent cast (SC) film, bi-layered solvent cast/3D printed (SC/3D) carrier and solvent cast/nanofiber (SC/NF) carrier.

**Figure 9 pharmaceutics-11-00678-f009:**
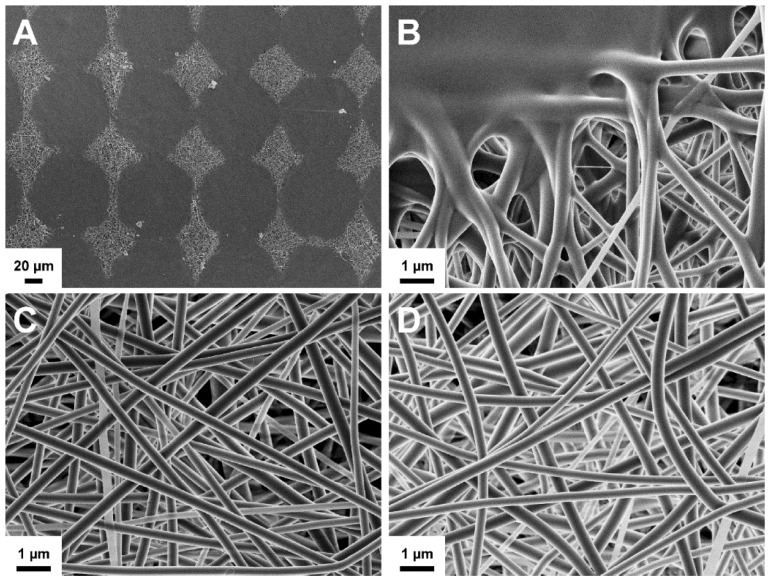
Scanning electron microscopy (SEM) images of non-crosslinked solvent cast/nanofiber (SC/NF) carrier after inkjet printing (**A**,**B**), and crosslinked SC/NF carrier before (**C**) and after (**D**) inkjet printing.

**Figure 10 pharmaceutics-11-00678-f010:**
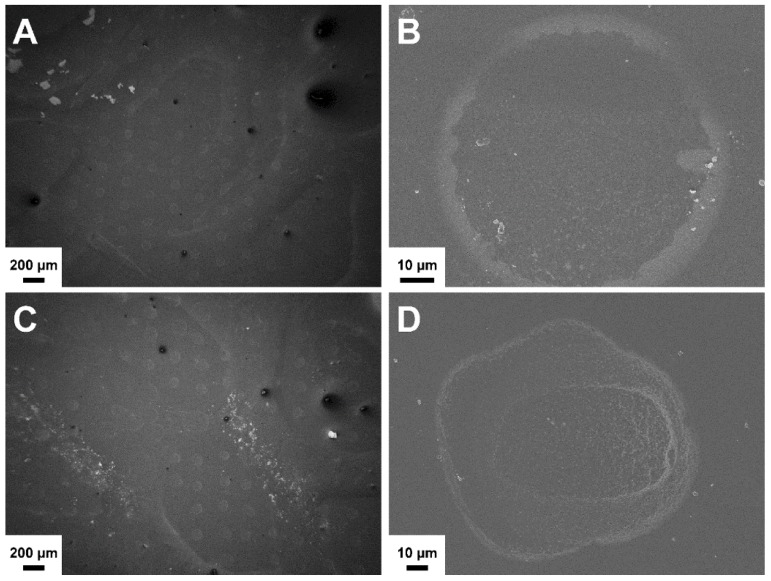
Scanning electron microscopy (SEM) images of non-crosslinked (**A**,**B**) and crosslinked (**C**,**D**) solvent cast/3D printed (SC/3D) carriers after inkjet printing.

**Table 1 pharmaceutics-11-00678-t001:** Physical properties of the solvent cast (SC) film, the electrospun nanofiber (NF) mat, the patterned 3D printed (3D) mat and the bi-layered carriers.

Sample Treatment	Thickness ^a^ (mm)	Puncture Test ^a^		Swelling Degree ^b^ (%) After 24 h	Degradation Degree ^b^ (%) After 7 h
		Burst Strength (N)	Distance at Break (mm)		
**Solvent Cast (SC) Film**					
non-crosslinked	0.04 ± 0.01	40.2 ± 14.9	5.0 ± 2.1	NA	NA
crosslinked	0.03 ± 0.02	37.4 ± 30.2	3.1 ± 0.9	↑ 69 ± 19	↓ 1.2 ± 1.2
**Electrospun Nanofiber (NF) Mat**					
non-crosslinked	NA	NA	NA	NA	NA
crosslinked	NA	NA	NA	↑ 401 ± 52	↓ 4.9 ± 5.9 ^c^
**Patterned 3D Printed (3D) Mat**					
non-crosslinked	0.04 ± 0.01	5.3 ± 1.1	2.9 ± 0.7	NA	NA
crosslinked	0.05 ± 0.01	5.3 ± 1.3	2.6 ± 0.4	↑ 75 ± 39	↓ 6.7 ± 0.7
**Bi-Layered Solvent Cast/Nanofiber (SC/NF) Carrier**					
non-crosslinked	0.05 ± 0.01	36.8 ± 5.7	6.4 ± 0.8	NA	NA
crosslinked	0.09 ± 0.02	45.5 ± 12.4	3.4 ± 0.8	↑ 338 ± 35	↓ 0.2 ± 0.3
**Bi-Layered Solvent Cast /3D Printed (SC/3D) Carrier**					
non-crosslinked	0.08 ± 0.03	38.4 ± 31.2	4.3 ± 1.8	NA	NA
crosslinked	0.11 ± 0.04	31.3 ± 10.9	2.4 ± 0.4	↑ 133 ± 20	↓ 2.8 ± 0.6

^a^ mean ± standard deviation, *n* = 10; ^b^ mean ± standard deviation, *n* = 3; ^c^ after 24 h; NA—not applicable.

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
