# Peer review of "Bi-Layered Polymer Carriers with Surface Modification by Electrospinning for Potential Wound Care Applications"

_pharmaceutics, 2019, doi:10.3390/pharmaceutics11120678_

Round 1

Reviewer 1 Report

The article entitled, "Bi-layered polymer carriers with surface modification by electrospinning for wound care applications" by Miraj et al., describes the preparation of three different forms of hydrogel polymer and compared their physico-chemical and mechanical properties. Though the authors measured relevant properties that are important as a wound dressing materials, it is important to measure the biocompatibility interms of cell viability of of the dressings for skin cells prior to loading of any antimicrobial/anti-inflammatory components. Considering quality of the journal, it is important to carry out at this stage rather than in an another manuscript. The following are additional minor comments to improve the quality of their work. 

On Page 5, line# 25 and Page 6, line #39 the authors mentioned Figure A1 and A2 but I am not able to find these Figures in the manuscript. 

SEM cross-section of the SC/3D and SC/NF along with SC films should be provided to discern the differences between various dressings.

The mechanism of crosslinking and the concomitant increase in crystallinity of SC and NF samples need to be explained and correlated. There appears a weak band around 1720 cm-1 after crosslinking of NF or SC samples. This need to be assigned and explained.   

Cytotoxicity and cell proliferative properties of the dressings in the crosslinked form for one of the skin cells/cell lines must be presented.   

Author Response

Reviewer 1:

Point 1: The article entitled, "Bi-layered polymer carriers with surface modification by electrospinning for wound care applications" by Miraj et al., describes the preparation of three different forms of hydrogel polymer and compared their physico-chemical and mechanical properties. Though the authors measured relevant properties that are important as a wound dressing materials, it is important to measure the biocompatibility interms of cell viability of of the dressings for skin cells prior to loading of any antimicrobial/anti-inflammatory components. Considering quality of the journal, it is important to carry out at this stage rather than in an another manuscript. The following are additional minor comments to improve the quality of their work.

On Page 5, line# 25 and Page 6, line #39 the authors mentioned Figure A1 and A2 but I am not able to find these Figures in the manuscript.

Response 1: Figures A1 and A2 are presented in the Supplementary A. This additional information has been added to the manuscript as follows:

Pages 5-6, lines 224-226: The SC films obtained were transparent with smooth surface (Supplementary Figure A1). After thermal cross-linking the films turned to yellowish, but no other structural changes nor cracking were observed (Supplementary Figure A1).

Page 6, lines 238-239: The 3D printed pattern was clearly visible after printing (Supplementary Figure A2) and the dried mats remained intact after removal from the copier film.

Point 2: SEM cross-section of the SC/3D and SC/NF along with SC films should be provided to discern the differences between various dressings.

Response 2: The visualization of the cross-sections would be also of interest to the authors. Unfortunately, the cross-sections of the bi-layered carriers and the solvent cast films were not visualized by scanning electron microcopy (SEM) due to the unavailability of an appropriate tool for preparing the cross-sections. It is important to prepare the samples without any modification; however, our trials which were conducted in order to get nice cross-section failed presumably due to the material behavior (fibers were smashed together) within the cutting places. Freeze fracture has been used to prepare cross-sections of hollow fiber and co-electrospun microfibers for visualization using SEM [1]. However, it has been shown that dropping fibers in liquid nitrogen can result in significant mechanical deformation in samples [1]. There is a practical upper limit to the size of the sample that can be prepared (i.e. it has to be no more than a few millimeters thick) [1] which also may cause problems in some cases. It has been reported that there is a considerable effect on fiber misalignment on the measurement results and thus corrections should be performed on sample geometry [2]. As this required a separate study on its own, this was not conducted within the present study.

Zhou, F.-L.; Parker, G.J.M.; Eichhorn, S.J.; Hubbard Cristinacce, P.L. Production and cross-sectional characterization of aligned co-electrospun hollow microfibrous bulk assemblies. Mater. Charact. 2015, 109, 25–35. Lorbach, C.; Hirn, U.; Kritzinger, J.; Bauer, W. Automated 3D measurement of fiber cross section morphology in handsheets. Nord. Pulp Pap. Res. J. 2012, 27, 264–269.

Point 3: The mechanism of crosslinking and the concomitant increase in crystallinity of SC and NF samples need to be explained and correlated. There appears a weak band around 1720 cm-1 after crosslinking of NF or SC samples. This need to be assigned and explained.   

Response 3: Additional information related to the spectral changes pointed out by the Reviewer has been added to manuscript (Section 3.4).

Point 4: Cytotoxicity and cell proliferative properties of the dressings in the crosslinked form for one of the skin cells/cell lines must be presented.

Response 4: The authors are aware of the lack of biologically relevant tests in the manuscript. The aim of the current work was mainly to investigate the preparation methods of bi-layered carriers and evaluate the physicochemical aspects of these prepared carriers. These studies provided evidence that it is relevant to move forward with more biorelevant biological tests. The cell studies, including cytotoxicity and cell migration/proliferation studies, are aimed to be addressed thoroughly in a follow-up project. We believe that only in vitro cytotoxicity testing of the materials safety is not enough and may be also not as relevant as additional living cell-mat interaction tests, since all materials used are FDA approved materials. It is very important point and all risk assessment and safety testing as well cell migration/proliferation tests (artificial wound models) will be conducted in further development. We have modified the manuscript and highlighted that more biorelevant tests are needed and will be conducted in future projects.

Reviewer 2 Report

Major concern

The major concern with the manuscript is lack of biological evidence to the effectiveness and applicability of the developed wound dressing. The bioadhesion was tested on an artificial skin which practically only represents the epidermis. The bioadhesion on epidermis and dermis/hypodermis is very different and cannot be correlated. No cell studies were performed. There are several simple wound animal models that can be used for this study including punch biopsy model.

Other concerns:

The work is complex but the rationale and motivation for such bilayering is not clear - electrospinning vs 3D printing? The physical characteristics for monolayers - solvent cast layer, electrospun layer, and 3D printed layer alone should have been provided. I know it is more work but they are very important controls. It would be beneficial to load a drug into the wound dressings to further provide evidence for "bioactive wound dressings".

Author Response

Reviewer 2:

Point 1: The major concern with the manuscript is lack of biological evidence to the effectiveness and applicability of the developed wound dressing. The bioadhesion was tested on an artificial skin which practically only represents the epidermis. The bioadhesion on epidermis and dermis/hypodermis is very different and cannot be correlated. No cell studies were performed. There are several simple wound animal models that can be used for this study including punch biopsy model.

Response 1: The authors thank the reviewer for excellent suggestions for testing the bioadhesion in much more biorelevant conditions. These methods were not available to the authors during the time of this project. However, future collaborations and/or experiments at the local facilities will be considered in the future. The authors are highly aware of the lack of biologically relevant tests in the manuscript. The bioadhesion and cell studies are aimed to be carried out in a follow-up project that would focus specifically on these issues.

Other concerns:

Point 2: The work is complex but the rationale and motivation for such bilayering is not clear - electrospinning vs 3D printing?

Response 2: The aim was to investigate the effect of surface modification on the physical properties and (bio)adhesiveness of the carriers. The different surface structures were obtained by the mentioned methods (electrospinning and 3D printing). The results indicate that the electrospinning method seems to be more suitable for bi-layered carriers prepared from a polymer mixture of polyvinyl alcohol and sodium alginate.

Point 3: The physical characteristics for monolayers - solvent cast layer, electrospun layer, and 3D printed layer alone should have been provided. I know it is more work but they are very important controls.

Response 3: The available physical characteristics of these formulations have been added to Table 1 in the manuscript. The discussion on the changes to the physical characteristics in the bi-layered carriers due to the addition of a second layer on the solvent cast film has been presented previously.

Point 4: It would be beneficial to load a drug into the wound dressings to further provide evidence for "bioactive wound dressings".

Response 4: The authors thank the reviewer for this obvious remark. The addition of drug(s)/bioactive material(s) is planned to be investigated in the follow-up project.

Reviewer 3 Report

The reviewed manuscript entitled: “Bi-layered Polymer Carriers with Surface Modification by Electrospinning for Wound Care Applications” by Palo et. al. could be regarded as a fine piece of original research with good balance of fundamental and application-oriented input. It is authors virtue citing the most relevant published results in the field and especially citation [27] - the work of Maver et. al..which seems being inspirational to the current research as carboxymethyl cellulose is close relevant to the natural alginate carbohydrate polymer. The choice of the polymeric pair PVA and SA is reasonable.

I have only few questions and critical remarks as follow:

1.      The thermal cross-linking of the PVA/SA blends is mostly explained in terms of physical cross-linking e.g. inter/intra-molecular H-bond formation and structure stabilization. Do your instrumental analysis detect any chemical cross-linking e.g. carboxylic/hydroxyl groups interactions and ester bond formation or the sodium form of COO:- restrict such possibility?

2.       What do you think could happen if you treat the polymeric PVA/SA blend with dilute acid solution where sodium alginate will turn to free alginic acid form which is known to be gel-forming and also in the prospect of possible in situ inter/intra-molecular polymer-polymer complex formation based on H-bonding between the carboxylic acid groups and the hydroxylic one? Have you tested such stabilization approach or ionic cross-linking by Ca2+ ions for example?

Author Response

Reviewer 3:

The reviewed manuscript entitled: “Bi-layered Polymer Carriers with Surface Modification by Electrospinning for Wound Care Applications” by Palo et. al. could be regarded as a fine piece of original research with good balance of fundamental and application-oriented input. It is authors virtue citing the most relevant published results in the field and especially citation [27] - the work of Maver et. al..which seems being inspirational to the current research as carboxymethyl cellulose is close relevant to the natural alginate carbohydrate polymer. The choice of the polymeric pair PVA and SA is reasonable.

I have only few questions and critical remarks as follow:

Point 1: The thermal cross-linking of the PVA/SA blends is mostly explained in terms of physical cross-linking e.g. inter/intra-molecular H-bond formation and structure stabilization. Do your instrumental analysis detect any chemical cross-linking e.g. carboxylic/hydroxyl groups interactions and ester bond formation or the sodium form of COO:- restrict such possibility?

Response 1: A way to detect chemical crosslinking in terms of ester bond formation has been suggested for example in a polyvinyl alcohol/glutaraldehyde hydrogels by Mansur et al. (reference [62]). Analysis of the spectra showed some changes in the spectral range (1750 – 1700 cm-1) suggested. However, the underlying reasons remain uncertain. Additionally, spectral changes were observed for the intensity of the absorbance band of OH- groups that also relates to the water content in the formulations. Short note on the topic has been added to the manuscript (Section 3.4).

Point 2: What do you think could happen if you treat the polymeric PVA/SA blend with dilute acid solution where sodium alginate will turn to free alginic acid form which is known to be gel-forming and also in the prospect of possible in situ inter/intra-molecular polymer-polymer complex formation based on H-bonding between the carboxylic acid groups and the hydroxylic one? Have you tested such stabilization approach or ionic cross-linking by Ca2+ ions for example?

Response 2: The pre-tests were performed with electrospun nanofiber (NF) mats by applying ionic (CaCl2), ultraviolet (UV) and thermal (150 °C and 180 °C) crosslinking methods. These pre-test results were not included in the current manuscript in order to avoid adding complexity to the study and to provide emphasis on the suitable crosslinking method. The thermal crosslinking at 180 °C for 10 min showed to be the most suitable out of the methods tested. However, the highest limit of the extent of crosslinking was not determined. Indeed, the ionic crosslinking provided electrospun materials which existence in aqueous conditions was prolonged the most. The ionic crosslinking with Ca2+ was shown to be unsuitable due to the extensive shrinking of the electrospun mat and the changes in the surface structure.

Round 2

Reviewer 1 Report

As I mentioned in my previous email, cytotoxicity must be carried out at this stage. Considering the scope and quality of the journal I think it is unfair to leave it out. I do not concur with the author's comments that cytotoxicity will be communicated as a separate work.  

Author Response

As requested by the Reviewer, the cytotoxicity and cell behavior on the substrates in the crosslinked form for BHK-21 cell line is presented. We have modified the manuscript and added all relevant proofs (Figures, Appendix Figure and modified Graphical abstract) from our biological tests. In addition, we have confirmed the viability of cells in the presence of our carrier materials both in crosslinked and non-crosslinked form. We conducted the living cell-carrier interaction tests using MTS testing in order to understand whether the different surface modification changes the cell behavior. All materials were safe, and no statistically significant differences were observed between the carriers (thermally crosslinked and non-crosslinked). Crosslinked bi-layered SC/NF carrier was the best surface supporting the cell viability the most. The cell viability on SC film and SC/3D carriers was approximately the same.

Reviewer 2 Report

I authors have stated that they are unable to perform additional studies as mentioned in reviewer's Comment 1 and Comment 4. These analyses are essential for the study and cannot be deferred/postponed for future.

Author Response

Response 1: As requested by the Reviewer, the cytotoxicity and cell behavior on the substrates in the crosslinked form for BHK-21 cell line is presented. We have modified the manuscript and added all relevant proofs (Figures, Appendix Figure and modified Graphical abstract) from our biological tests. In addition, we have confirmed the viability of cells in the presence of our carrier materials both in crosslinked and non-crosslinked form. We conducted the living cell-carrier interaction tests using MTS testing in order to understand whether the different surface modification changes the cell behavior. All materials were safe, and no statistically significant differences were observed between the carriers (thermally crosslinked and non-crosslinked). Crosslinked bi-layered SC/NF carrier was the best surface supporting the cell viability the most. The cell viability on SC film and SC/3D carriers was approximately the same.

Drug incorporation into these bi-layered carriers or on these carriers will be evaluated in the future studies, as this is a logical continuation of the present work.

Round 3

Reviewer 1 Report

The authors reported the mammalian cell viability data seeded onto various matrices. This preliminary information is important and sufficient to demonstrate the biocompatibility  of the prepared materials in vitro. The manuscript now can be accepted and I congratulate the authors for their work.   

Author Response

Dear Reviewer,

Thank you for your time and useful comments for the improvement of the manuscript.

Best wishes,

Reviewer 2 Report

The manuscript has significantly improved over two revisions and can now be accepted. I suggest the title to be modified to include "potential" for wound dressing applications as:

Bi-layered polymer carriers with surface modification by electrospinning for potential wound care applications.

Author Response

Dear Reviewer,

Thank you for your useful comments which helped to improve the manuscript. We agree with the suggestion to add one more word to the title.

Best wishes,